# Turning challenges into opportunities: Lessons from Ethiopia's COVID-19 response for strengthening health systems and health security

Muluneh Yigzaw Mossie[1]*, Jenny Shannon[2], Amare Bayeh Desta[3], Etsub Brhanesilassie[4], Adugna Dufera[5], Saira Nawaz[2]

1 PATH, Ethiopia Country Office, Addis Ababa, Ethiopia, 2 PATH, Seattle, Washington, United States of America, 3 CDC Foundation, Addis Ababa, Ethiopia, 4 USAID Ethiopia, Addis Ababa, Ethiopia, 5 Ethiopian Public Health Institute, Addis Ababa, Ethiopia

* yigzawm@gmail.com

## Abstract

Emerging in late 2019, the COVID-19 outbreak rapidly evolved into an unprecedented global public health crisis. While exposing critical health system weaknesses, the pandemic also spurred innovation and investment, including rapid vaccine development. This paper examines Ethiopia's multi-tiered health system response, exploring the strategies, enabling conditions, barriers, and lessons learned. We conducted a mixed-methods, cross-sectional study across all tiers of Ethiopia's health system from August to December 2023. Data came from three concurrent components: a quantitative survey of implementers (n = 952), key-informant interviews (n = 80), and administrative data. An additional 233 health workers were surveyed to assess intervention integration. Study tools were informed by the Consolidated Framework for Implementation Research and WHO pillars. Quantitative data were analyzed using descriptive statistics, while qualitative data underwent framework analysis, followed by synthesis and triangulation. Results show Ethiopia rapidly expanded laboratory capacity, increasing RT-PCR sites from one (February 2020) to 72 (June 2020), and developed genomic sequencing. Critical care capacity grew, supported by the acquisition of over 764 oxygen concentrators, 320 mechanical ventilators, and remote workforce training. Rapid vaccine deployment and cold chain expansion facilitated over 68 million people receiving at least one dose. The country gained valuable experience in community engagement and deployed over 3,000 rapid response teams. While initial political will and engagement were strong, it waned over time. Consequently, despite efforts to integrate and institutionalize, some key interventions— including surveillance, laboratory testing, and case management—declined as the pandemic's urgent phase passed and political engagement diminished. In conclusion, the Ethiopian health system gained experiences and tools in critical care, laboratory testing, rapid vaccine deployment, and community engagement. If institutionalized

**Data availability statement:** The quantitative survey data supporting the findings of the study are submitted as an additional supplementary file. We have not shared the administrative data because our agreement with the Ministry of Health restricts its use solely to this study. The data can be requested directly from the Ministry of Health of Ethiopia (info@moh.gov.et), and the authors are happy to facilitate any such requests.

**Funding:** This study was funded by the United States Agency for International Development (Award #: 72066322CA00003). The funding supported the design, implementation, and analysis of the study. However, the content of this manuscript is solely the responsibility of the authors and does not necessarily reflect the views of the United States Agency for International Development or the United States Government.

**Competing interests:** The authors have declared that no competing interests exist.

and sustained, these capacities can significantly strengthen Ethiopia's health system and enhance its resilience. To fully integrate and institutionalize these platforms for future emergencies, robust monitoring is essential.

## Introduction

Emerging in late 2019, the COVID-19 outbreak rapidly evolved into an unprecedented global public health crisis [1–4]. By April 2023, when the World Health Organization (WHO) declared it no longer a public health emergency of international concern, the virus had infected over 765 million people and claimed more than 6.9 million lives globally [5]. While the pandemic exposed critical weaknesses in health systems and global coordination, it also spurred innovation and renewed investment in public health infrastructure, including the rapid development and deployment of vaccines [6].

To support national responses, WHO introduced a strategic framework encompassing eleven key response pillars, including coordination, risk communication, surveillance, laboratory testing, and case management [7]. These pillars align closely with core components of primary health care and broader health system functions, highlighting the potential of pandemic response strategies to enhance long-term system strengthening efforts.

In Ethiopia, the first COVID-19 case was reported in March 2020 [8]. Despite resource constraints and limited preparedness [9], the government mounted a comprehensive response aligned with WHO guidance. These efforts included establishing multi-sectoral coordination, fostering community engagement, expanding surveillance and testing, improving case management, and launching national vaccination efforts [10]. A national state of emergency was declared, followed by a series of public health and social measures, including school closures, restrictions on gatherings, suspension of religious services, travel limitations, and quarantine protocols. Public awareness campaigns were also launched to promote preventive practices. These interventions aimed to delay virus transmission, reduce strain on the health system, and buy time to scale up the country's preparedness and response capacity [11].

This study was conducted in Ethiopia shortly after the WHO lifted the global emergency status for COVID-19. It aimed to examine how the country implemented its response across various levels of the health system, exploring the strategies adopted, enabling conditions, barriers encountered, and lessons learned. Particular attention is given to how key interventions were integrated into and sustained by the broader primary health care system. Through this analysis, we aim to contribute to the global evidence base on how health systems can operate and adapt during large-scale public health crises, and leverage these insights to strengthen health systems and health security.

## Methods

We conducted a mixed-methods, cross-sectional study across all tiers of Ethiopia's health system. Data came from three concurrent components: a quantitative survey,

key-informant interviews, and administrative data. The survey quantified the prevalence and perceived importance of implementation factors (contributors and barriers), while the key informant interviews added rich narratives on challenges, lessons learned and successes. Administrative data offered objective indicators and trends that anchored the other findings, such as changes in the availability of laboratories, critical care facilities, and vaccines.

## Sampling techniques

We adapted the sampling strategy proposed by Peter et al. [12] to identify populations of actors involved in complex and multi-institutional initiatives. This approach consisted of three key steps: identifying the target population or population universe, defining the source population, and drawing samples from the source population. In this context, the target population was defined as all individuals meeting the inclusion criteria, while the source population referred to the practically accessible segment of this target population who could be feasibly reached for data collection. Hence, the target population in Ethiopia comprised individuals who had been directly involved in the implementation of COVID-19-related activities for a continuous period of 12 months or more between January 2020 and April 2023. Implementing activities refer to all aspects of the COVID-19 response, including coordination, resource mobilization, risk communication, surveillance, laboratory testing, case management, logistics and supply chain management, infection prevention and control, continuity of essential services, research, data management, and vaccination efforts.

Sampling was stratified by both the type of activity conducted and the geographic location to ensure representation across all pillars of the COVID-19 response and the major contextual settings of the country: urban, agrarian, and pastoralist areas. The country was first stratified into these three contexts, from which five regions were randomly selected. Within each selected region, institutions leading the COVID-19 response were identified. From these institutions, all individuals involved in implementing the response who met the inclusion criteria were listed. Individuals were then randomly selected from these lists for a quantitative survey. In cases where only one eligible individual was available, that individual was included (Fig 1). A subset of these quantitative survey participants was then purposively selected for key informant interviews.

## Sample size

Our quantitative survey employed a complex sampling design, which included adjustments for the design effect and potential non-response, to estimate the minimum sample size. This estimation resulted in the inclusion of 952 eligible individuals in the study. Of these, 80 participants were also selected for key informant interviews, with their number determined by data saturation. Additionally, 233 health workers were enrolled from health facilities to specifically assess the implementation of key pandemic response interventions within these facilities and examine their integration into the broader health system.

## Data collection

We developed and tested three data collection tools: a structured questionnaire for the quantitative survey, a separate structured questionnaire for individuals working in health facilities, and a semi-structured interview guide for key informant interviews. All tools were developed informed by the Consolidated Framework for Implementation Research (CFIR) domains and designed to cover all WHO pillars of the COVID-19 response. Following tool development, enumerators were recruited and received five days of training, which included pilot testing of the tools outside the study areas. Most interviews were conducted face-to-face; however, a few were conducted virtually via video calls. Virtual interviews were arranged for key individuals who played major roles in the response, were included in the sample, but were unavailable for face-to-face interviews during the data collection period. To ensure data accuracy, all key informant interviews were audio-recorded and transcribed verbatim. Data collection was carried out between August and December 2023.

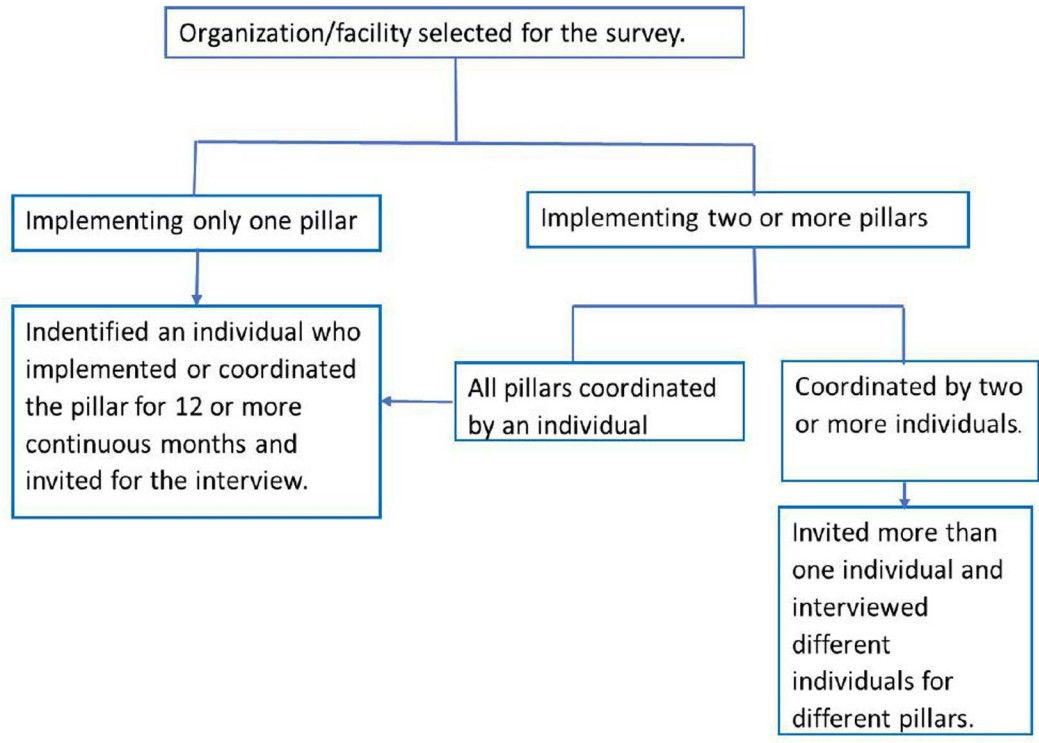

**Fig 1. Study participant selection process within institutions.**

## Analysis

Data analysis was conducted at three levels: quantitative analysis, qualitative analysis, and synthesis and triangulation. The quantitative survey data underwent descriptive analysis, identifying barriers and facilitators to the implementation of COVID-19 response interventions. This analysis also assessed the level of adoption of guidelines and practices at the health facility level. Guided by CFIR domains and constructs, the analysis examined determinants across the health system, political, economic, social, and technological environments, incorporating both outer and inner CFIR settings. Additionally, trends in key service indicators were computed from the administrative data.

For the qualitative component—including key informant interviews and document reviews—an initial codebook was developed based on the study objectives, the WHO COVID-19 Monitoring and Evaluation Framework, and the interview guide. The codebook was refined following a preliminary review of KII transcripts. Two independent coders conducted detailed line-by-line coding. A framework analysis approach was applied, using both deductive and inductive methods: predefined themes guided the deductive process, while the inductive approach allowed new insights to emerge directly from the data. Atlas.ti software facilitated systematic organization and retrieval of coded content.

Finally, findings from the quantitative and qualitative strands were triangulated to build a comprehensive understanding of the barriers, facilitators, successes, and lessons learned. Data from all sources were synthesized to enhance interpretation and insight. The robustness of the evidence supporting key findings was assessed across three dimensions: the number of data sources (single vs. multiple), the type of data (perception-based vs. fact-based), and the overall quality of the data.

**Global Public Health**

## Ethical statement

The Ethiopian Public Health Institute Institutional Review Board (IRB) approved the study (Protocol Number: EPHI-IRB-512–2023). Recruitment of study participants took place from August 7, 2023, to December 15, 2023, in accordance with ethical guidelines. As per IRB approval, verbal consent was obtained from all participants after they received and read the consent form, which detailed the study's purpose, potential benefits, risks, and possible harms. Participants provided verbal agreement to proceed with the interview, and the interviewer documented consent by marking a designated section on the form. All completed consent forms were securely retained to ensure proper ethical documentation and compliance. Additionally, all data were deidentified and stored on a secured, password-protected device to maintain participant confidentiality and data security.

## Results

Table 1 presents an overview of study respondents' engagement across various COVID-19 pillars from January 2020 to April 2023. More than one out of two (56.0%) respondents participated in the Vaccination pillar. This was followed by Risk Communication and Community Engagement (RCCE) (52.0%), and Infection Prevention and Control (49.5%).

The government of Ethiopia established a range of directives and regulatory frameworks to facilitate the implementation of the COVID-19 response interventions, particularly during the initial phase of the pandemic, and was able to step-up its efforts to slow down the spread of the virus. The political environment was cited as a key contributor to Ethiopia's COVID-19 response. As shown in Table 2, respondents identified a conducive political environment as a contributing factor in maintaining essential health services (64.7%), managing points of entry (63.0%), RCCE (62.0%), IPC (61.8%), vaccination (60.4%), surveillance and outbreak investigation (60.4%), case management (56.2%), logistics and supply systems (52.2%), and laboratory testing (46.2%).

This finding was supported by respondents from key informant interviews. A key informant from the Ethiopian Public Health Institute (EPHI), for instance, described the government's role in establishing legal frameworks as a key contributing factor for the entire response:

*"From the outset, laws, rules, and regulations were issued frequently. Policies were introduced requiring individuals to maintain physical distance, wash their hands, and ensure that hotels, schools, and other crowded service facilities operated in compliance with these regulations. Adjustments were made as necessary based on evolving circumstances. If the government had been unable to establish such regulations, implementing the [COVID-19 response] interventions would have been difficult."*

—Respondent, EPHI

**Table 1. Distribution of study participants by COVID-19 pillar.**

| Pillar | Number | Percent |
| --- | --- | --- |
| Surveillance | 396 | 41.6 |
| Laboratory testing | 208 | 21.8 |
| Case Management | 210 | 22.1 |
| IPC | 471 | 49.5 |
| RCCE | 495 | 52.0 |
| Logistics and supply systems | 324 | 34.0 |
| Vaccination | 533 | 56.0 |
| Points of entry | 62 | 6.5 |
| Maintaining essential services | 238 | 25.0 |

IPC = Infection prevention and control; RCCE = Risk communication and community engagement.

**Table 2. Distribution of respondents citing political will as the most significant contributor to COVID-19 Response.**

| WHO Pillar | Number | Percent |
|---|---|---|
| Surveillance (n = 396) | 239 | 60.4 |
| Laboratory testing (n = 208) | 96 | 46.2 |
| Case management (n = 210) | 118 | 56.2 |
| IPC (n = 471) | 291 | 61.8 |
| Vaccination (n = 533) | 322 | 60.4 |
| RCCE (n = 495) | 307 | 62.0 |
| Logistics and supply systems (n = 324) | 169 | 52.2 |
| Point of entry (n = 62) | 39 | 63.9 |
| Maintaining essential health services (n = 238) | 154 | 64.7 |

## Multi-sectoral and multi-stakeholder collaboration and action

A multi-sectoral and multi-stakeholder coordination mechanism was established at the first phase of the COVID-19 pandemic with higher political engagement. The task force, which was established through this coordination mechanism, rapidly devised new directives, outlining the dos and don'ts of non-pharmacological interventions, such as social and physical distancing, contact tracing, isolation and quarantine, use of personal protective equipment, restriction of movement, using mask and hand hygiene, and the actions related to international travel. A key informant from the Ministry of Health noted:

> *"Many stakeholders were involved in the COVID-19 coordination. These included the EPHI team, the ministry of health team, and other stakeholders. When the response plan was prepared, we involved not only the public health institute, but also various other stakeholders."*

> —Respondent, Ministry of Health

There were challenges in the coordination mechanisms specifically at the early phase of the pandemic. Conflict compromised the implementation of the COVID-19 national and sub-national responses, as the public failed to honor directives and government officials were unable to proactively engage in the task force as they prioritized other public demands. Furthermore, following the war in northern Ethiopia, the government's focus shifted to security, resulting in little engagement in the COVID-19 response. In the Amhara region, a respondent specifically highlighted the significant impact of the war on the COVID-19 response efforts within the region.

> *"...There was a conflict in northern Ethiopia…and during that time, high-level officials were seen chatting with military personnel in crowded conditions. People observed them without face masks in a meeting hall filled with many individuals. Consequently, the public misperceived that COVID-19 was not present in the country."*

> —Respondent, regional public health institute

Despite challenges and setbacks in the coordination mechanism, the Ministry of Health and its partners developed strategies to contain the pandemic, even in conflict-affected settings, with a particular focus on revitalizing vaccination efforts. Coordination and partnerships among stakeholders enabled the vaccination of notable number of people, even in these challenging areas. A key informant at sub-national level said:

> *"…to reach hard to reach areas, we have been using a mobile approach, where a mobile team is assigned to reach each cluster. We tried to serve IDP sites and refugee camps using this method. The funds required to establish the*

*mobile teams were found by coordinating different partners. There are also, hard to reach places …, places not reachable due to security reasons, where we followed a hit and run approach to reach them."*

—Respondent, zonal health department

**Training and deployment of rapid response teams**

Soon after the World Health Organization declared COVID-19 a public health emergency of international concern, the Ethiopian Ministry of Health established and revitalized RRTs. These teams were composed of epidemiologists, clinicians and WASH experts, risk communication and community engagement experts, laboratory personnel, and logisticians. The roles of RRTs were wide-ranging, covering nearly all aspects of the response. Vaccination, however, was managed by the Expanded Program on Immunization (EPI) unit at the Ministry of Health, independently of RRTs. Stationed at the national and regional Emergency Operating Centers (EOCs), RRTs were involved in preparing suspected cases for testing, facilitating their isolation when necessary, and ensuring their admission to case management centers as appropriate. A key informant described the roles of RRTs as:

*"...The primary component [of the COVID-19 response activities] is what we refer to as the rapid response team, which was deployed in all kebeles, the lowest level of administration in the country. So, the rapid response team was not solely focused on response activities but also conducted searches for suspected cases."*

—Respondent, Ministry of Health

**Contact tracing, quarantine and isolation**

In the initial phase of the pandemic, all suspects and travelers were quarantined until confirmed negative. Those found positive for the virus were also isolated regardless of the severity, although later home-based management became common practice rather than isolation in designated centers. Quarantining and isolation of an enormous number of people–all contacts, travelers, and positive cases– created a challenge in a health system that was already resource constrained. Inadequate supplies at quarantine and isolation centers hampered the capacity to safely isolate and manage all these suspected and confirmed cases. In addition, contact tracing was challenging due to deficits in communication skills among the RRTs as their training in interpersonal communications was limited due to an emergency deployment, hindering their ability to effectively engage with contacts. Adding to this, inadequate resources at isolation centers exacerbated hesitancy among suspected cases to cooperate and adhere to isolation and quarantine measures. A key informant remarked:

*"Managing a large number of people … proved to be quite demanding. Despite the efforts of various partners to supply food and water, meeting the needs of all individuals was challenging. Moreover, the warm weather conditions contributed to some individuals escaping from the quarantine center and rejoining society."*

—Respondent, zonal health department

To overcome resource constraints, the federal and regional governments undertook resource mobilization efforts, largely by engaging local communities, the private sector, and donors. Higher learning institutions also contributed by providing student facilities for the quarantine of returnees. A key informant from one of the regions noted:

*"The private sector contributed a lot in funding the [COVID-19] response, mostly in kind. We conducted meetings with industry managers and wealthy people living in the town. …After that, local businesses supported us with a lot of*

*money. … The textile industry produced many masks. Private health facilities collected money and brought us thousands of masks."*

—Respondent, regional health bureau

## Availability and quality of data for surveillance

When the COVID-19 pandemic struck, Ethiopia lacked reliable databases for surveillance and outbreak investigation. Surveillance personnel resorted to manual recording of cases, followed by the transfer of data to higher levels using available open applications. This approach resulted in errors in both recording and reporting, contributing to incomplete data. In addition, reporting materials ran out of stock, forcing surveillance personnel to search for copies while also dealing with time-consuming and complex paper forms. However, as the pandemic progressed into mid-2020, a notable milestone was the development of an integrated application suite. This application system incorporated surveillance, laboratory testing, point of entry, case management, and logistics information systems. Leveraging the DHIS2 platform, the application integrated all pillars and established interfaces to facilitate communication and data driven decision making among all responders involved in the COVID-19 response efforts. Two key informants explained the initial challenges and the role digitalization played in the COVID-19 response:

*"There was a system called DHIS2 for [real time recording of] COVID-19 that allowed us to register directly on the system. And we have access to the data from there…for this reason, the reports were immediate and no delay. The application allows professionals at zone and woreda [district] levels to analyze the data and generate information for action."*

—Respondent, regional health bureau

*Initially, the surveillance and data capture system relied on Excel, leading to issues with missing, inconsistent, and untimely data. However, the Ministry of Health leadership later prioritized digitalization, resulting in a fully digital system with integrated data interfaces, streamlining data from various sources.*

—Respondent, EPHI

## Surveillance at point of entry

Ethiopia established a network of 27 points of entry to monitor and control the influx of travelers during the COVID-19 pandemic. Respondents believe that despite being absent at an early phase of the pandemic, the deployment of surge teams at points of entry boosted response capabilities, allowing for rapid actions in the event of a surge in arrivals or suspected cases. A key informant at subnational level described:

*"We had dispatched a surge team to the area [point of entry] to conduct screening for infectious diseases, including COVID-19. It's important to note that the displaced individuals arriving from the conflict in Sudan included not only Ethiopians but also individuals of other nationalities."*

— Respondent, regional health bureau

Although Ethiopia set up a network of point-of-entry surveillance systems, porous borders and limited cross-border collaboration remained significant challenges. The absence of cross-border surveillance mechanisms further impeded effective outbreak investigations. In areas lacking formal checkpoints, people crossed the borders unchecked, mingling with nearby communities and breaching prevention guidelines.

## COVID-19 laboratory testing

When the COVID-19 pandemic occurred, Ethiopia's laboratory testing capacity was limited. In February 2020, only one laboratory in the country was capable of performing COVID-19 RT-PCR testing. Moreover, the absence of automation and the prolonged turnaround times for test results presented challenges during the early stages of the pandemic. Two respondents explained the state of laboratory services when the pandemic occurred in Ethiopia:

> *"Initially, we only had one molecular laboratory at a national level, located at EPHI, the Influenza lab. We didn't even have a molecular laboratory professional."*

—Respondent, Ministry of Health

> *"Initially, due to outdated methods and a limited number of professionals, it took over 24 hours to process samples. Moreover, with all samples from different regions being handled by a small team at the national lab, accessibility to laboratory tests was impacted."*

—Respondent, EPHI

Despite these initial setbacks, as the pandemic evolved, Ethiopia was able to expand testing centers. The country conducted an inventory of available PCR machines, including those in research centers across sectors like agriculture and higher education. Collaborative platforms were established with these organizations, enabling the repurposing of existing machines and enhancing testing capacity. Moreover, private laboratories in Addis Ababa played a role in expanding testing facilities by offering COVID-19 RT-PCR testing, particularly for travelers. Consequently, the number of laboratories equipped with RT-PCR testing capacity increased from just one in February 2020 to seventy-two by June 2020 (Fig 2).

In addition, Ethiopia was able to conduct in-country viral genome sequencing, enabling the country to monitor evolution of the virus. A key informant noted the following:

> *"…Initially, we didn't have a capacity for viral genome sequencing. So, tracking the variants was a challenge. I think it's still a challenge, but at least we now have in-country capacity. This capability is available at the national level, primarily at EPHI and a few other laboratories. So, identifying which variant is responsible for a surge in a specific area can now be determined locally. That is a significant development."*

—Respondent, Ministry of Health

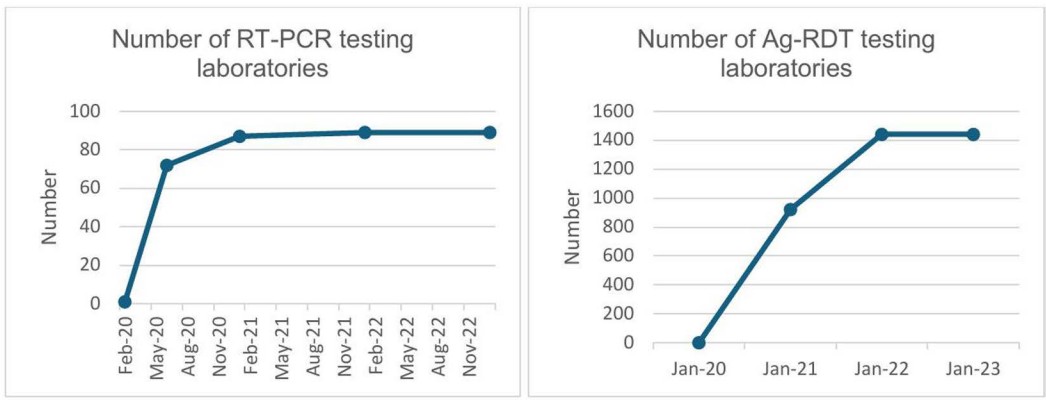

**Fig 2. Trends in COVID-19 laboratory testing capacity during the pandemic period.**

In the middle of the pandemic, Ethiopia approved and deployed antigen-based rapid diagnostic tests (Ag-RDTs) in areas where RT-PCR testing was not available. By 2022, 1,442 health facilities were conducting Ag-RDT testing for COVID-19. Over time, the automation of laboratory processes further enhanced daily testing capacity. The advent of rapid diagnostic tests and expansion of RT-PCR testing centers improved access to laboratory testing. In 2021, close to 78% of case management and isolation centers had standard operating procedures (SOPs) to conduct onsite COVID-19 testing, while it steadily declined there after (Fig 3).

Although the sharp increase of testing center for COVID-19, the infrastructure of laboratories remained unchanged, resulting in insufficient space for testing activities. This led to overcrowding and congestion within the laboratories,

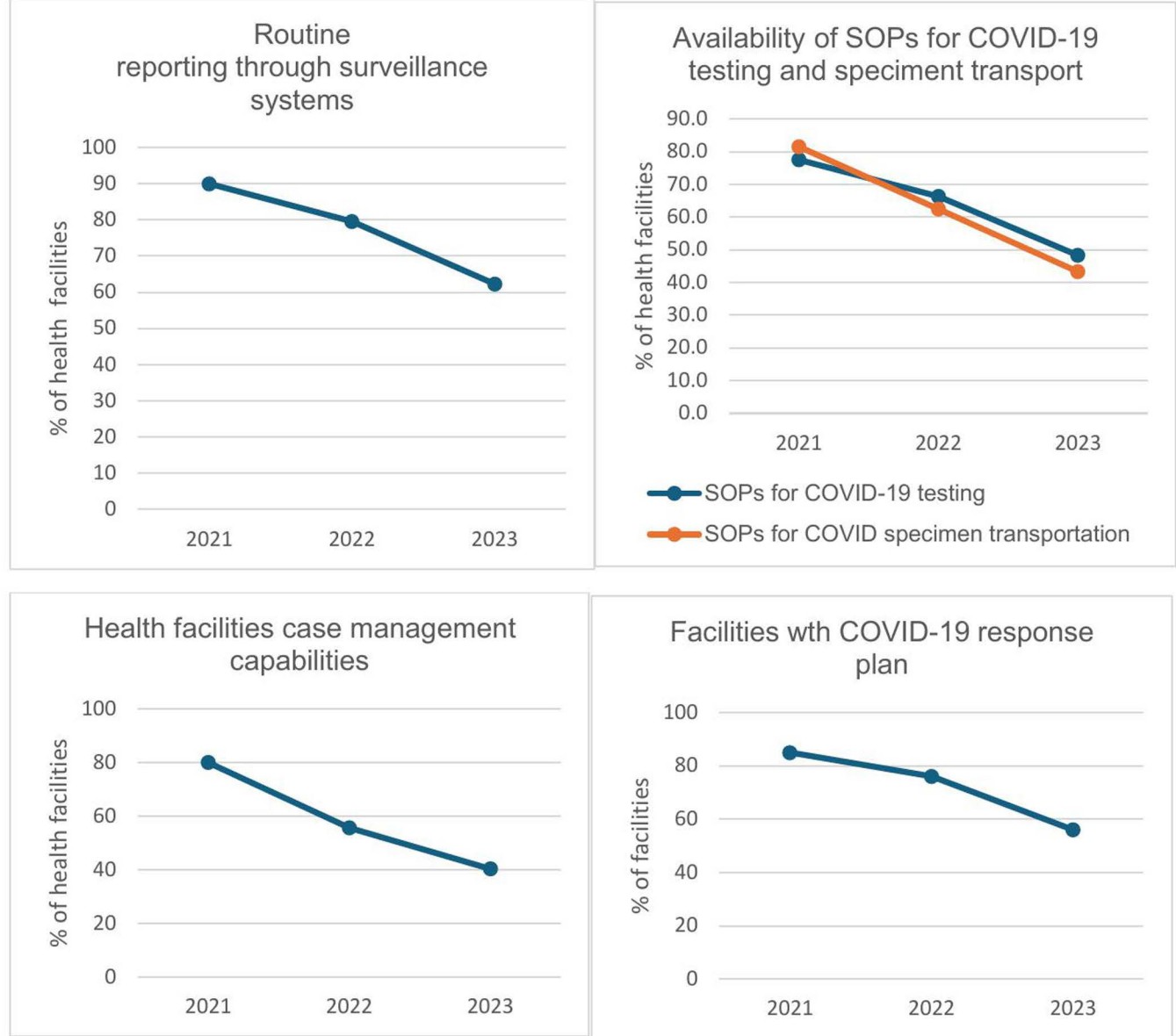

**Fig 3. Trends in implementing key intervention in health facilities from 2021 to 2023.**

increasing the risk of sample mixing and contamination. A key informant described the limited spaces available for laboratory testing and the challenges of quality assurance of laboratory testing:

> "…the infrastructure in the laboratories did not meet the standard for a molecular lab. … Some regions simply provided one room and expected us to work there. However, this was difficult because contamination is a significant challenge in molecular samples and could even result in false positive results"
>
> —Respondent, EPHI

In addition, ensuring access for all populations to laboratory services remained challenging due to Ethiopia's vast geography and the growing number of COVID-19 cases. To address access issues in remote areas without testing centers, samples were collected and transported using existing infrastructure. Ethiopia's COVID-19 testing strategy utilized the country's established systems and experiences for sample collection, transport, and testing, which were already in place for diseases such as tuberculosis, HIV, and sentinel surveillance of Severe Acute Respiratory Illness and Influenza-like illnesses.

**Case management**

When the COVID-19 pandemic hit, Ethiopia had limited capacity to manage severe COVID-19 cases, as unanimously noted by all key informants. The country was grappling with a lack of trained health providers, ventilators and other essential equipment and supplies for critical care, although case management centers were established early in the pandemic. To address this, an online training platform was established, and several health providers were trained remotely.

> "We organized several training sessions on COVID-19 for healthcare workers at the early phase of the pandemic, which served as a crucial starting point since it was a new disease and there was limited knowledge on how to respond. To address this, we established an online training platform that enabled us to train many health workers in a short period. We also created a central information repository, compiling the latest guidance and publications about COVID-19. This repository was made accessible through a dedicated platform."
>
> — Respondent, professional associations

By the end of 2021, the number of case management centers established nationwide reached 155. With the expansion of case management centers, the number of critical care devices also grew. Throughout the pandemic, Ethiopia secured over 764 oxygen concentrators and 320 mechanical ventilators (Fig 4).

A key informant from the Ministry of Health described the improvements in critical care facilities as follows:

> "…the case management has changed a lot of things, for example, our oxygen system has been changed, beds were not connected to oxygen in any of the hospitals before the COVD pandemic. In addition to the oxygen plant, oxygen cylinders have been procured in large quantities. Mechanical ventilators were few before COVID, but now we have many."
>
> —Respondent, Ministry of Health

As case management centers expanded, the inadequate availability and unsuitability of personal protective equipment (PPE) jeopardized the safety and well-being of frontline health workers, increasing their risk of contracting the virus while treating patients. This shortage also heightened the risk of COVID-19 transmission within health facility premises, further disrupting essential health services in both case management centers and other facilities. In the later stages of the

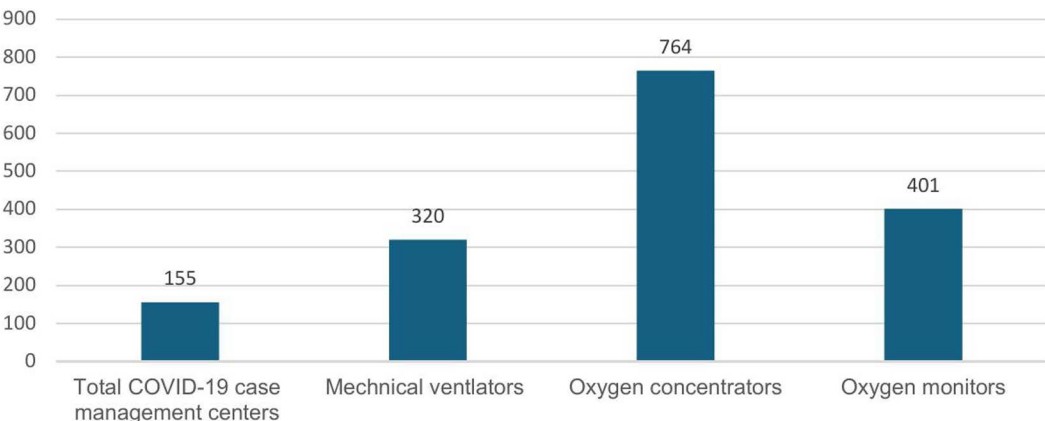

**Fig 4. Total number of COVID-19 case management centers, mechanical ventilators, oxygen concentrators and oxygen monitors secured during the pandemic.**

pandemic, improvements in PPE availability and usage, along with increased awareness among health workers and the public, led to a reduction in disruptions to essential services (Fig 5).

## Logistics and supply systems

A subcommittee under the national COVID-19 task force was established to coordinate logistics and supply availability, focusing initially on items such as PPE, mechanical ventilators, laboratory devices, and, later, vaccines. This committee, composed of members from the Ministry of Health and other sectors, was instrumental in ensuring efficient resource distribution.

Initially, as the pandemic created global demand, widespread shortages of supplies emerged in international markets. The persistent scarcity and rising costs of PPE imposed financial burdens on both the government and the public in Ethiopia. In response, the logistics and supply chain task force explored local production options, collaborating with local companies with manufacturing experience. Through these public-private partnership efforts, local production capabilities were rapidly mobilized, leading to the production of essential supplies within a month and at affordable prices. A key informant explained how development of local production capacities improved cost and availability of face masks:

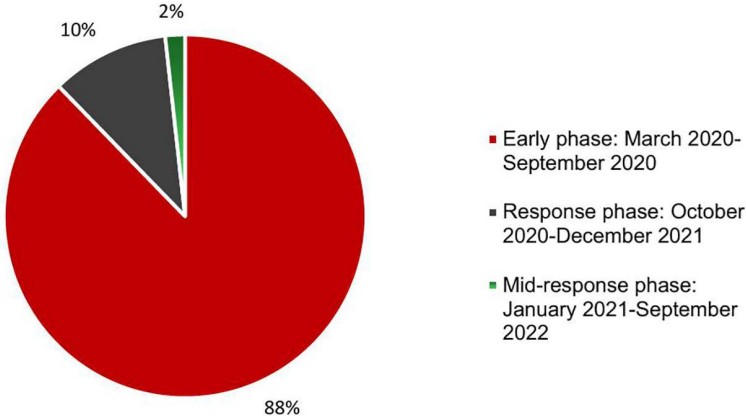

- Early phase: March 2020-September 2020
- Response phase: October 2020-December 2021
- Mid-response phase: January 2021-September 2022

**Fig 5. The phase of the COVID-19 pandemic when the most disruption to the health system was observed.**

*"During the early stages of the pandemic, the price of a single face mask skyrocketed from its usual 5 birr to as high as 350 birr. However, within a month or even less, we successfully ramped up production and imports of masks, consequently stabilizing the market. This achievement directly resulted from the strategies we put in place and stands as one of the measures, among others, that contributed to our improved control over the pandemic."*

—Respondent, Ministry of Health

Coordination and collaboration among sectors were instrumental in expediting the last mile delivery of logistics. By working closely with the Ministry of Finance and Customs, health authorities at the Ministry of Health and regional health bureaus were able to enhance the distribution and timely access to essential items required for the response. New directives were also established to facilitate the distribution of essential supplies to final destinations within a short period. A key informant described the role of multi-sectoral coordination in the swift transportation and delivery of essential supplies:

*"The coordination across sectors made the logistics to go out quickly in a short period of time and the items were not warehoused… In the past, …commodities imported as emergency used to stay at customs for 15 days and even months, but during the time of COVID-19 pandemic, the commodities arrived [to final destinations] immediately. The Ministry of Finance, Customs, EFDA working together, especially the laws issued by the EFDA, made the COVID-19 response commodities reach the final destination within a short period."*

—Respondent, Ethiopian Pharmaceutical Supply Services

### Vaccination

The commencement of the COVID-19 vaccination initiative in Ethiopia occurred exactly one year after the country's first reported COVID-19 case. COVID-19 vaccination started within the established EPI structures and coordination platforms, leveraging their existing functions. These included utilizing pre-existing advisory groups and committees like the Inter-agency Coordination Committee (ICC) and the National Immunization Technical Advisory Group (NITAG).

Ethiopia aimed to vaccinate up to 20% of the target population in the first phase of its vaccination initiative [13]. However, the country only managed to vaccinate 3% of the target within six months. While limited vaccine stock was a challenge, an ineffective vaccination strategy was also another barrier at the first stage of the vaccination drive. When the initial approach failed to reach target populations, Ethiopia adopted a hybrid delivery strategy with campaign style approach playing a central role, allowing the program to access underserved groups and substantially increasing the number of people receiving vaccine doses (Fig 6). Beyond increasing vaccine uptake, this approach offered opportunities to integrate other services, such as identifying and reducing the number of zero-dose children.

Moreover, Ethiopia enhanced its cold chain infrastructure, particularly the ultracold chain capacity. With support from development partners, ultracold chain facilities were established in all regions of the country, significantly increasing their availability.

### Service integration

The Ministry of Health developed guidelines and has begun integrating COVID-19 response activities and strategies into the broader health system since 2022. According to these guidelines, all health facilities were expected to incorporate COVID-19 measures into routine services, as the disease had transitioned to an endemic phase. The plan called for integrating laboratory testing with existing services and aligning COVID-19 surveillance with established disease surveillance protocols. Additionally, a national effort documented best practices for sustained use beyond the outbreak. However, these best practices were not done at lower levels. Furthermore, our analysis revealed a steady decline in

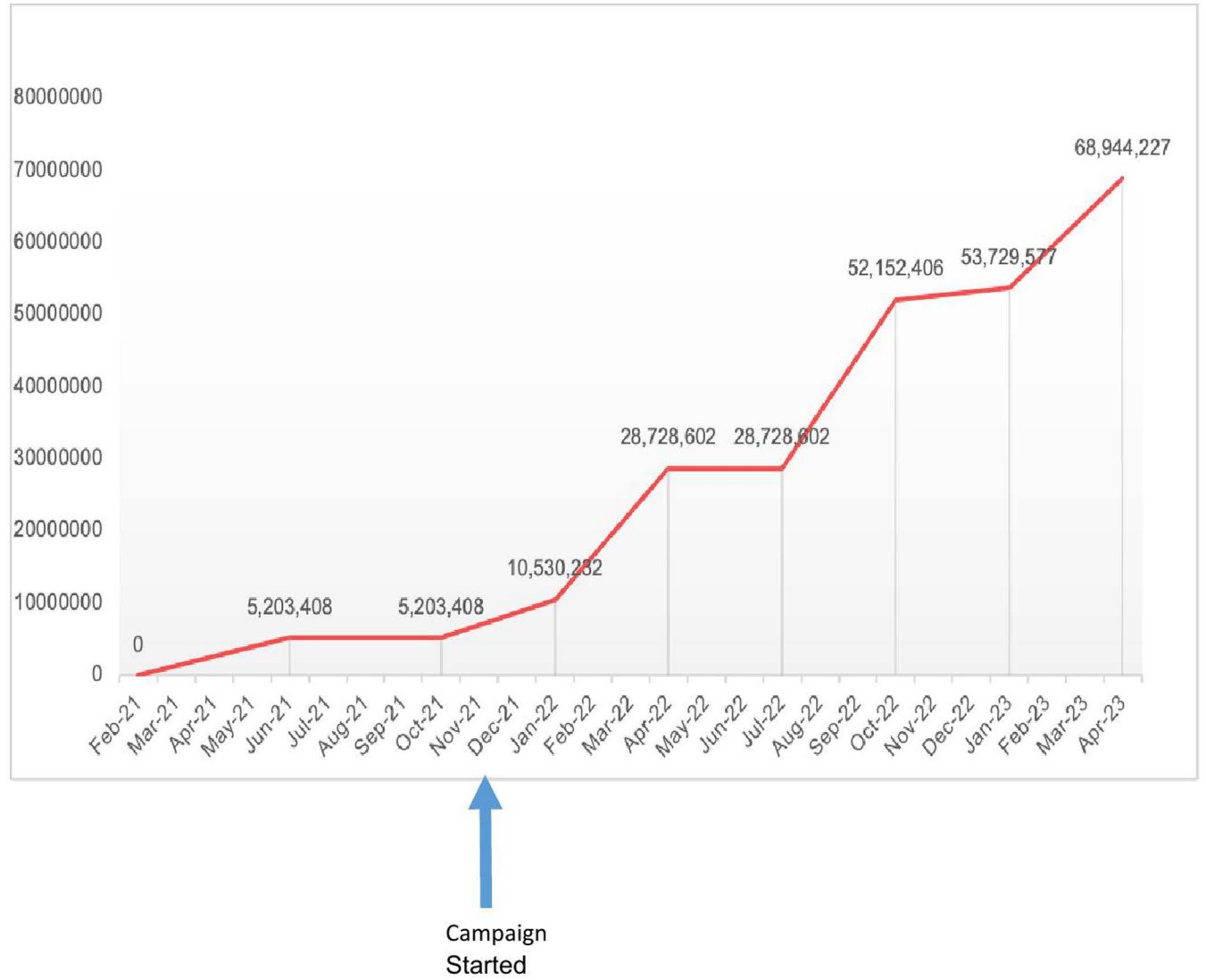

**Fig 6. Trends in Covid-19 Vaccine Doses Administered in Ethiopia.**

COVID-19-related activities within health facilities even prior to 2023, when COVID-19 was still considered a public health emergency of international concern. As shown in Fig 3, the percentage of facilities equipped with functional lifesaving devices and supplies to manage COVID-19 cases dropped from 79.9% in 2021 to 40.3% in 2023. Similarly, the proportion of facilities with SOPs for laboratory testing and specimen transport for COVID-19 declined from 81.5% in 2021 to 43.3% in 2023.

## Discussion

Ethiopia implemented a comprehensive set of interventions to contain the COVID-19 pandemic, despite facing numerous challenges. Key measures included multi-sectoral collaboration and action, point-of-entry screening and quarantine, active case finding and isolation, community engagement, expansion of testing and case management facilities, and later, vaccination efforts. With strong political commitment from the outset, Ethiopia swiftly engaged various sectors and mobilized

resources and personnel. Early in the outbreak, the government established 27 points of entry equipped with staff and facilities to screen and quarantine travelers, aiming to prevent the importation and spread of the virus.

At the onset of the pandemic, Ethiopia had limited capacity for laboratory testing and managing severe COVID-19 cases [9]. When the WHO declared COVID-19 a public health emergency of international concern on 30 January 2020, Ethiopia lacked laboratories capable of conducting SARS-Cov-2 RT-PCR tests and had insufficient life-saving equipment to manage severe cases. This was a common challenge among many low- and middle-income countries (LMICs), where health systems were generally weak [14]. In these settings, laboratory testing capacity was low, and critical care facilities were limited in the ability to provide adequate care for severe COVID-19 cases [15].

Despite Ethiopia's limited laboratory testing and case management capacity, the early implementation of non-pharmaceutical interventions effectively delayed the peak of COVID-19 transmission. This delay provided critical time to expand testing capacity and prepare for severe cases, all without imposing strict lockdowns. Although the first COVID-19 case was reported in mid-March 2020, the peak was delayed for several months, occurring between late August and early September 2020 [10]. Similar experiences were reported in other countries. Robust point-of-entry measures, along with vigilant surveillance, social distancing, handwashing and case isolation [16], effectively delayed pandemic spread, buying time to scale up case management and laboratory services. Perhaps, the role of these interventions in infectious disease prevention and outbreak containment is well-established [17,18] and are foundational to primary health care systems today. This aligns with historical public health principles. Long before antibiotics were discovered, the germ theory transformed disease prevention, highlighting the importance of hygiene and quarantine measures, which were also key strategies in managing COVID-19.

Across all pillars, Ethiopia's COVID-19 response leveraged public-private partnerships and collaboration with various sectors, civil society, donors, and international partners for resource mobilization, planning, and funding. Crucially, collaboration between the agricultural sector, higher learning institutions, and the health sector went beyond traditional boundaries, enabling shared resources and access to critical services. Notably, the health sector's collaboration with universities and agricultural research institutions facilitated the rapid scaling up of laboratory testing, as PCR machines from these institutions were repurposed for COVID-19 testing. Furthermore, universities provided student facilities for quarantine and isolation, highlighting the significant role higher learning institutions played during the pandemic. The importance of multi-sectoral and multi-stakeholder collaboration in health is well-documented and forms a core pillar of primary health care systems [19]. Furthermore, public-private partnerships are increasingly recognized for their role in health system development, as shared resources and collaborative efforts help secure essential supplies and foster innovation [20]. The private sector played a pivotal role in Ethiopia's COVID-19 response, particularly by expanding access to laboratory testing, availing PPE, and other essential facilities, showcasing the important role the private sector can play in public health interventions.

In Ethiopia's COVID-19 response, public figures played a role in strengthening community trust and promoting adherence to non-pharmaceutical interventions, particularly physical and social distancing. The involvement of religious leaders and other prominent figures helped build community confidence during a time when misinformation and disinformation were widespread, encouraged broader public engagement, and supported compliance with response guidelines and protocols. Building trust and fostering effective public relations have long been recognized as essential for managing health emergencies and other public health interventions, particularly in primary health care systems [21–23]. Reflecting this, global primary health care guidelines emphasize community engagement, as highlighted in a recent WHO and UNICEF publication on the PHC operational framework [19]. Moreover, evidence shows that those countries effectively controlled COVID-19 without strict lockdowns partly relied on strong community trust, underscoring its critical role in disease prevention and outbreak response [16].

A key pillar of the COVID-19 response was the provision of essential supplies, including PPE and, later, vaccines. During the pandemic, scarcity of these items was a challenge. Shortages of these essential items were partly due to

global supply chain disruptions and vaccine nationalism [24,25], and partly due to inefficient supply systems which are a characteristic of weak health systems [26]. Global supply disruptions were overcome by expanding local production capacity in Ethiopia.

The provision of essential medical supplies is recognized as a foundational pillar in both the WHO's health system building blocks [27] and the primary health care framework [19]. The availability of such critical supplies is essential for the everyday resilience of health systems as well as in mitigating outbreaks. However, LMICs often face persistent gaps in these supplies. While insufficient financing is a contributing factor, inefficiencies in supply chains—particularly in last-mile delivery—pose significant challenges [26]. The global supply chain issues highlighted by COVID-19 emphasize the need to develop local capacity for producing essential medical supplies. By building local production capacity through public-private partnerships, the availability of essential supplies and self-sufficiency can be improved, contributing to stronger primary health care systems.

Manual data recording initially affected data quality, particularly for COVID-19 surveillance and data-driven decision-making in Ethiopia. However, collaborative efforts improved digitalization across the response, streamlining data for surveillance, laboratory testing, and case management. This rapid digital transformation highlighted the potential of digital health solutions in resource-limited settings and has implications for strengthening health systems. Together with previous initiatives, such as the Information Revolution Roadmap [28,29], this experience offers a valuable learning opportunity for the Ethiopian government to advance its health information systems.

## Implications for health system strengthening

Ethiopia has extensive experience in primary healthcare, largely centered on a community health program led by health extension workers. However, recent studies highlight challenges stemming from fragmentation within the program [30]. Launched in 2003, Ethiopia's community health program was initially designed to empower communities and families in managing their own health, serving as an entry point into the health system [31]. Over time, this focus on community empowerment has diminished as shifting priorities diverted attention from this core objective, leading to fragmented efforts [30,32]. While Ethiopia's National Health Sector Development Program (HSDP), implemented from the late 1990s to 2015, recognized the community health program led by health extension workers as a flagship component of the health system [30], it notably did not place significant emphasis on involving public figures and other informal community structures as part of its community engagement and trust-building strategy. Instead, community engagement was facilitated through the one-to-five network and the Health Development Army approaches [33], which were not explicitly utilized during the COVID-19 response.

The distinct role of community engagement and multi-sectoral action observed in Ethiopia's initial COVID-19 response, particularly the involvement of public figures, underscores the value of adapting community engagement strategies for both primary health care and health security. While the Ministry of Health has recently taken steps to optimize the existing community health program [34], valuable lessons from the COVID-19 response remain to be fully integrated. Specifically, engaging communities through public figures and revitalizing multi-sectoral action enhances the implementation of health programs [35,36], builds public trust, and, when combined with other measures, supports the realization of strong primary health care systems.

The COVID-19 response offers other valuable opportunities for the primary health care system in Ethiopia to learn and grow. One key takeaway is the training provided to health workers across various aspects of the response, primarily through an online platform. This platform has established a legacy, serving as a resource for continuous professional development in Ethiopia's health system. Furthermore, the over 3,000 RRTs deployed during the pandemic gained extensive experience in managing various outbreak response activities. If properly documented, these RRTs could be rapidly mobilized in the event of future emergencies, representing a significant asset for the development of the health workforce.

The other positive legacy of the COVID-19 response in Ethiopia includes significant developments in laboratory testing, critical care, and digitalization. The expansion of molecular laboratories and the establishment of viral genome sequencing capabilities enhance the country's ability to detect and monitor diseases. In addition, the health system secured critical medical devices and deployed ultracold chain facilities nationwide. Combined with experiences in digitalization, these developments, if integrated into routine healthcare practices, could transform Ethiopia's health system and ensure resilience.

## The need for an integrated health system framework

Over the past decade and a half, Ethiopia's health system strengthening efforts have focused on WHO's six building blocks, while health security efforts have centered on IHR core capacities, often overlooking the role of health system strengthening in health security. An additional emphasis on primary health care—though widely recognized as integral to health system development—has further fragmented decision-making, policy, and resource allocation. Some policymakers emphasize health security, downplaying system strengthening, while others champion primary health care as central to health system development [37]. This divergence raises a key question when considering how to integrate COVID-19 response lessons into the broader health system: should these lessons be incorporated into health system strengthening via the six WHO building blocks, the IHR core capacities, primary health care pillars, or all these areas? To harness COVID-19 response lessons and strengthen health systems, Ethiopia needs an integrated, context-informed health system framework that reduces siloed approaches, as the WHO building blocks may not fully address health security needs. As Blanchet et al. noted, a health system's resilience is its "capacity to absorb, adapt and transform when exposed to a shock such as a pandemic, natural disaster or armed conflict and still retain the same control over its structure and functions" [38]. Integrating lessons from the COVID-19 response into an adapted framework would enable Ethiopia's health system to fulfill these characteristics and better mitigate future emergencies.

Our analysis showed that while Ethiopia attempted to integrate COVID-19 response experiences into its broader primary health care system for routine practice, the implementation of some of the recommended practices declined even before the pandemic ended. This decline highlights the need for leadership that prioritizes service integration into routine practices, as well as a unified framework to guide implementation and ensure resilience. Without such a framework, these lessons and experiences may not be fully sustained, as system level integration has not materialized, and decision making is fragmented. With fragmented decision-making, oversight and monitoring of these valuable lessons within the broader health system will be limited, hindering efforts to ensure everyday resilience.

## Limitations

There were a few limitations to this study. First, the health facilities selected for the sample were deliberately chosen from those designated as case management and isolation centers, with the purpose of obtaining comprehensive information about the response efforts. Consequently, the findings from these facilities may not fully represent all health facilities across the country. Second, the retrospective nature of the information sought may cause inaccurate recall, which could affect the study's findings. Despite these limitations, the study offers valuable insights into COVID-19 response efforts, with data triangulated from multiple sources to enhance data quality and validity.

## Conclusion

The Ethiopian health system gained valuable experiences and tools during the COVID-19 response, particularly in health workforce development, critical care, digital data solutions, cold chain systems, point of entry, laboratory testing, and community engagement. If these capacities are integrated and institutionalized into the broader health system—particularly within primary health care—and sustained, they have the potential to significantly strengthen the health system, enhance its resilience in daily operations, and provide a robust foundation for preventing and mitigating future risks. However, we observed a decline in some practices, particularly in surveillance, laboratory testing, and case management after the initial

urgency waned. Our findings thus suggest that while COVID-19 investments expanded life-saving interventions, robust monitoring and follow-up mechanisms are essential to fully integrate and institutionalize many COVID-19 response platforms for future emergencies.

## Supporting information

**S1 File. Reported COVID 19 Response Barriers.**
(DOCX)

**S2 File. Facility survey data used in the analysis.**
(CSV)

**S3 File. Quantitative survey data used in the analysis.**
(CSV)

**S4 File. Individuals survey questionnaire.**
(DOCX)

**S5 File. Health Facility Questionnaire.**
(DOCX)

**S6 File. Key informant interview guide.**
(DOCX)

**S1 Checklist. Inclusivity-in-global-research-questionnaire.**
(DOCX)

## Author contributions

**Conceptualization:** Muluneh Yigzaw Mossie, Jenny Shannon, Amare Bayeh Desta, Etsub Brhanesilassie, Saira Nawaz.

**Data curation:** Muluneh Yigzaw Mossie, Jenny Shannon, Saira Nawaz.

**Formal analysis:** Muluneh Yigzaw Mossie, Jenny Shannon, Saira Nawaz.

**Funding acquisition:** Saira Nawaz.

**Investigation:** Muluneh Yigzaw Mossie, Amare Bayeh Desta, Etsub Brhanesilassie, Adugna Dufera, Saira Nawaz.

**Methodology:** Muluneh Yigzaw Mossie, Jenny Shannon, Amare Bayeh Desta, Etsub Brhanesilassie, Adugna Dufera, Saira Nawaz.

**Project administration:** Muluneh Yigzaw Mossie, Amare Bayeh Desta.

**Resources:** Muluneh Yigzaw Mossie, Saira Nawaz.

**Software:** Muluneh Yigzaw Mossie, Jenny Shannon, Saira Nawaz.

**Supervision:** Muluneh Yigzaw Mossie, Amare Bayeh Desta, Etsub Brhanesilassie, Adugna Dufera, Saira Nawaz.

**Validation:** Muluneh Yigzaw Mossie, Jenny Shannon, Amare Bayeh Desta, Etsub Brhanesilassie, Adugna Dufera, Saira Nawaz.

**Visualization:** Muluneh Yigzaw Mossie, Jenny Shannon, Adugna Dufera, Saira Nawaz.

**Writing – original draft:** Muluneh Yigzaw Mossie.

**Writing – review & editing:** Muluneh Yigzaw Mossie, Jenny Shannon, Amare Bayeh Desta, Etsub Brhanesilassie, Adugna Dufera, Saira Nawaz.

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
