## [Decision Letter · Decision Letter 0]

6 May 2025

PGPH-D-25-00504

Turning Challenges into Opportunities: Lessons from Ethiopia’s COVID-19 Response for Strengthening Health Systems and Health Security

Dear Dr. Mossie,

Thank you for submitting your manuscript to PLOS Global Public Health. After careful consideration, we feel that it has merit but does not fully meet PLOS Global Public Health’s publication criteria as it currently stands. Therefore, we invite you to submit a revised version of the manuscript that addresses the points raised during the review process.

The manuscript has been evaluated by two reviewers, and their comments are available below.

Could you please carefully revise the manuscript to address all comments raised?

We look forward to receiving your revised manuscript.

Kind regards,

Jennifer Tucker, PhD

Staff Editor

Journal Requirements:

1. Please include a complete copy of PLOS’ questionnaire on inclusivity in global research in your revised manuscript. Our policy for research in this area aims to improve transparency in the reporting of research performed outside of researchers’ own country or community. The policy applies to researchers who have travelled to a different country to conduct research, research with Indigenous populations or their lands, and research on cultural artefacts. The questionnaire can also be requested at the journal’s discretion for any other submissions, even if these conditions are not met. Please find more information on the policy and a link to download a blank copy of the questionnaire here: https://journals.plos.org/globalpublichealth/s/best-practices-in-research-reporting. Please upload a completed version of your questionnaire as Supporting Information when you resubmit your manuscript. 2. Please send a completed 'Competing Interests' statement, including any COIs declared by your co-authors. If you have no competing interests to declare, please state "The authors have declared that no competing interests exist". Otherwise please declare all competing interests beginning with the statement "I have read the journal's policy and the authors of this manuscript have the following competing interests:" 3. Your current Financial Disclosure states, “This study was funded by the United States Agency for International Development (USAID). The funding supported the design, implementation, and analysis of the study. However, the content of this manuscript is solely the responsibility of the authors and does not necessarily reflect the views of USAID or the United States Government.”. However, your funding information on the submission form indicates that you received funding from “President's Malaria Initiative”. Please indicate by return email the full and correct funding information for your study and confirm the order in which funding contributions should appear. Please be sure to indicate whether the funders played any role in the study design, data collection and analysis, decision to publish, or preparation of the manuscript. 4. In the online submission form, you indicated that The data supporting the findings of this study are available upon request from the corresponding author. All PLOS journals now require all data underlying the findings described in their manuscript to be freely available to other researchers, either 1. In a public repository, 2. Within the manuscript itself, or 3. Uploaded as supplementary information. This policy applies to all data except where public deposition would breach compliance with the protocol approved by your research ethics board. If your data cannot be made publicly available for ethical or legal reasons (e.g., public availability would compromise patient privacy), please explain your reasons by return email and your exemption request will be escalated to the editor for approval. Your exemption request will be handled independently and will not hold up the peer review process, but will need to be resolved should your manuscript be accepted for publication. One of the Editorial team will then be in touch if there are any issues. 5. Please provide separate figure files in .tif or .eps format. For more information about figure files please see our guidelines:  https://journals.plos.org/globalpublichealth/s/figures https://journals.plos.org/globalpublichealth/s/figures#loc-file-requirements

Additional Editor Comments (if provided):

Reviewers' comments:

Reviewer's Responses to Questions

**Comments to the Author**

1. Does this manuscript meet PLOS Global Public Health’s publication criteria ? Is the manuscript technically sound, and do the data support the conclusions? The manuscript must describe methodologically and ethically rigorous research with conclusions that are appropriately drawn based on the data presented.

Reviewer #1: Yes

Reviewer #2: Partly

2. Has the statistical analysis been performed appropriately and rigorously?

Reviewer #1: N/A

Reviewer #2: No

3. Have the authors made all data underlying the findings in their manuscript fully available (please refer to the Data Availability Statement at the start of the manuscript PDF file)?

Reviewer #1: No

Reviewer #2: No

4. Is the manuscript presented in an intelligible fashion and written in standard English?

Reviewer #1: No

Reviewer #2: No

5. Review Comments to the Author

Reviewer #1: Manuscript Number: PGPH-D-25-00504

Manuscript Title: Turning Challenges into Opportunities: Lessons from Ethiopia’s COVID-19 Response for Strengthening Health Systems and Health Security

Dear Authors

Thank you for giving me the opportunity to review the paper “Turning Challenges into Opportunities: Lessons from Ethiopia’s COVID-19 Response for Strengthening Health Systems and Health Security; PGPH-D-25-00504”

General Comments

Overall, I think this is definitely a relevant paper. A lot of work has been published about the challenges to service delivery due to the COVID-19 pandemic. Little has been published about lessons for building back better so I think the work would add to the existing evidence on resilience and transformation in light of the COVID-19 pandemic.

Abstract:

1. Please make the abstract structured with clear titles for each of the sections:

2. From what I understood in the methods, you did not conduct any surveys but rather interviews with key informants.

Introduction

3. Could you please refer to the definition of resilience in this introduction. The gist of this manuscript is learning to transform or building back better. This is well explained in this paper: Blanchet K, Nam SL, Ramalingam B, Pozo-Martin F. Governance and Capacity to Manage Resilience of Health Systems: Towards a New Conceptual Framework. Int J Health Policy Manag. 2017 Aug 1;6(8):431-435. doi: 10.15171/ijhpm.2017.36. PMID: 28812842; PMCID: PMC5553211.

Methods:

4. Could you please state when the data were collected. This is important for us to know how sustained the lessons were from the interventions to respond to COVID-19.

5. Line 127 I think you meant study population and not study universe.

6. Could you please state that the selection of participants was purposive because you describe the rationale (purpose) of selecting the individuals who participated in the study.

7. Line 134 – What is EPHI? Please write in full at first mention.

8. I like the fact that the key informants included covered the broad strategic operations of an emergency response.

9. Please describe what a woreda is so that an international reader can relate.

10. What informed the design of the guides for the key informant interview guides? Any literature? Framework? Or theory?

11. The use of deductive and inductive approaches to analysis is a good approach to arrive at depth and breadth beyond prescribed frameworks and/ or theories.

12. The authors do not describe the quantitative data and how they were analyzed. Where were they obtained from? Which indicators were analyzed?

Results

13. I think the results describe many of the response strategies. I am curious as to how the system got better as a result of these response interventions. Could you provide some results that reflect “the system becoming stronger” to say provide vaccination services or maternal and child health services or any other services as a result of interventions during the response. The reason I make this comment is to show how indeed the system learned from the response.

14. I am curious as to the implementation of non-pharmaceutical interventions – movement restrictions, closure of entertainment and religious venues? Were these implemented? It would be good to highlight these as they affected service delivery and use.

Discussion:

15. Could you discuss how the quantitative results informed the qualitative results?

16. Could you also discuss whether/ if there were deliberate strategies to document lessons and learnings? This is important because it is these lessons that strengthen the health system to deliver other services. In addition, if there are deliberate efforts to document lessons, they are more likely to become part of routine service delivery as opposed to adhoc unplanned lessons.

17. I do not think recall bias as it is applied to epidemiology applies to this study. Recall bias is not forgetting.

Conclusions

18. Please do link the conclusions to the results

19. Please check the references and ensure they align with journal formatting guidance. Especially the abbreviation of Journal names in the reference.

Reviewer #2: Dear author,

Thank you for asking me to review this paper titled’ Turning Challenges into Opportunities: Lessons from Ethiopia’s COVID-19 Response for Strengthening Health Systems and Health Security’.

The authors need to carefully consider a better way to systematically write and present this paper with readers in mind so that they can have a clearer understanding of the content, and possible reproduction of the work in the future.

My suggestions and comments are listed below;

General comment

The method section was not properly written and the scientific content for a method section was inadequate and not detailed.

Abstract

1. Rationale for the study was not stated.

2. What is concurrent nested mixed methods design? There is no such design in Epidemiology. Authors should state the proper study design used

3. Process of analysis should be included in the abstract

Introduction

1. Rationale for the study was not stated.

Methods

The method section was not well written, or explained and not organised. It lacked structure. There is a format for writing method section, but the authors have not fully followed this, the section was muddled up. In addition, author should explain systematically, what data was collected from which group of people or where.

1. What is concurrent nested mixed methods design? There is no such design in Epidemiology. Authors should state the proper study design used. Mixed method approach is a data collection method combining both qualitative and quantitative methods of data collection.

2. Move lines 123b – 125

3. Study sample lines 126 – 156: this section is confusing and wrongly put together.

i. Authors should state the sampling technique used

ii. Then re-organise the sampling procedure according to the sampling technique used.

iii. Is this sample for quantitative or qualitative data collection?

4. Which quantitative data did author collect? Among which group of people? Was it secondary data analysis of records etc? These should be well stated in the methods section.

5. Authors should state source of questionnaire used for data collection.

6. How were the data generated?

7. What statistical analysis was done on the quantitative data? How were the charts/figure drawn? Trend analysis?

8. Authors should state the Key informant interview guide used for this study.

9. How was the KII guide developed?

10. How many KII was conducted?

11. What was the process of the interview? In terms of how many KII were conducted, how long did each interview take? How was the interview recorded?

It was difficult to follow through with the result and discussion sections because of the methodological flaws.

6. PLOS authors have the option to publish the peer review history of their article (what does this mean? ). If published, this will include your full peer review and any attached files.

**Do you want your identity to be public for this peer review?** For information about this choice, including consent withdrawal, please see our Privacy Policy .

Reviewer #1: **Yes: ** Steven N Kabwama

Reviewer #2: No

---

## [Decision Letter · Decision Letter 1]

28 Jul 2025

Turning Challenges into Opportunities: Lessons from Ethiopia’s COVID-19 Response for Strengthening Health Systems and Health Security

PGPH-D-25-00504R1

Dear Dr Mossie,

We are pleased to inform you that your manuscript 'Turning Challenges into Opportunities: Lessons from Ethiopia’s COVID-19 Response for Strengthening Health Systems and Health Security' has been provisionally accepted for publication in PLOS Global Public Health.

Best regards,

Julia Robinson

Executive Editor

Reviewer Comments (if any, and for reference):

Reviewer's Responses to Questions

**Comments to the Author**

1. If the authors have adequately addressed your comments raised in a previous round of review and you feel that this manuscript is now acceptable for publication, you may indicate that here to bypass the “Comments to the Author” section, enter your conflict of interest statement in the “Confidential to Editor” section, and submit your "Accept" recommendation.

Reviewer #1: All comments have been addressed

2. Does this manuscript meet PLOS Global Public Health’s publication criteria ? Is the manuscript technically sound, and do the data support the conclusions? The manuscript must describe methodologically and ethically rigorous research with conclusions that are appropriately drawn based on the data presented.

Reviewer #1: (No Response)

3. Has the statistical analysis been performed appropriately and rigorously?

Reviewer #1: (No Response)

4. Have the authors made all data underlying the findings in their manuscript fully available (please refer to the Data Availability Statement at the start of the manuscript PDF file)?

Reviewer #1: (No Response)

5. Is the manuscript presented in an intelligible fashion and written in standard English?

Reviewer #1: (No Response)

6. Review Comments to the Author

Reviewer #1: (No Response)

7. PLOS authors have the option to publish the peer review history of their article (what does this mean? ). If published, this will include your full peer review and any attached files.

**Do you want your identity to be public for this peer review?** For information about this choice, including consent withdrawal, please see our Privacy Policy .

Reviewer #1: **Yes: ** Steven N Kabwama
